# Electrostatic Ion-Acoustic Shock Waves in a Magnetized Degenerate Quantum Plasma

**Sharmin Jahan** [1,*], **Booshrat E. Sharmin** [1], **Nure Alam Chowdhury** [2], **Abdul Mannan** [1], **Tanu Shree Roy** [3] and **A A Mamun** [1]

1   Department of Physics, Jahangirnagar University, Dhaka 1342, Bangladesh; sharmin114phy@gmail.com (B.E.S.); abdulmannan@juniv.edu (A.M.); mamun_phys@juniv.edu (A.A.M.)
2   Plasma Physics Division, Atomic Energy Centre, Dhaka 1000, Bangladesh; nurealam1743phy@gmail.com
3   Department of Physics, Bangladesh University of Textiles, Tejgaon Industrial Area, Dhaka 1000, Bangladesh; tanu.jabi@gmail.com
*   Correspondence: jahan88phy@gmail.com

**Abstract:** A theoretical investigation has been carried out to examine the ion-acoustic shock waves (IASHWs) in a magnetized degenerate quantum plasma system containing inertialess ultra-relativistically degenerate electrons, and inertial non-relativistic positively charged heavy and light ions. The Burgers equation is derived by employing the reductive perturbation method. It can be seen that under the consideration of non-relativistic positively charged heavy and light ions, the plasma model only supports the positive electrostatic shock structure. It is also observed that the charge state and number density of the non-relativistic heavy and light ions enhance the amplitude of IASHWs, and the steepness of the shock profile is decreased with ion kinematic viscosity. The findings of our present investigation will be helpful in understanding the nonlinear propagation of IASHWs in white dwarfs and neutron stars.

**Keywords:** shock waves; Burgers equation; degenerate quantum plasma; reductive perturbation method

## 1. Introduction

The research regarding the propagation of nonlinear electrostatic excitations in a degenerate quantum plasma system (DQPS) has received a substantial attention from plasma physicists due to its ubiquitous existence in white dwarfs [1–3] and neutron stars [1–3]. It is believed that the components of the DQPS are electrons, positively charged heavy ions (e.g., $_{26}^{56}$Fe [4], $_{37}^{85}$Rb [5], $_{42}^{96}$Mo [5]), and positively charged light ions (e.g., $_{1}^{1}$H [6,7], $_{2}^{4}$He [8], $_{6}^{12}$C [9,10]). A number of authors investigated nonlinear waves in DQPS with positively charged heavy and light ions and electrons [11–14].

The characteristics of DQPS are comprehensively governed by the number density of the plasma species of DQPS, and it has been observed that the electron number density in white dwarfs is in the order of $10^{30}$ cm$^{-3}$ to $10^{39}$ cm$^{-3}$, and even more in neutron stars [9,10]. The dynamics of these high-dense plasma species in DQPS can be predicted by the Heisenberg uncertainty principle and Pauli exclusion principle, and under consideration of these two principles, the plasma species can create degenerate pressure which is readily outwardly directional and is not similar to the thermal pressure in normal plasmas. In extremely high-dense plasma, the degenerate pressure usually exceeds the thermal pressure. Therefore, the degenerate pressure has to be taken into account to model the dynamics of the DQPS. The degenerate pressure associated with degenerate electrons, heavy ions, and light ions can be given by [2]

$$P_s = \acute{K}_s N_s^\gamma, \tag{1}$$

where $s$ represents the electron or heavy ion or light ion species, i.e., $s = e$ for the electron species, $s = 1$ for the heavy ion species, and $s = 2$ for the light ion species;

$$\gamma = 5/3; \quad \acute{K}_s = (3/5)(\pi/3)^{\frac{1}{3}}(\pi\hbar^2)/m_s \simeq 3\Lambda_{cs}\hbar c/5, \qquad (2)$$

for the non-relativistic limit (with $\Lambda_{cs} = \pi\hbar/m_s c$, $\hbar$ is the Planck constant ($h$) divided by $2\pi$, $m_s$ is the mass of species $s$, and c is the speed of light in vacuum), and

$$\gamma = 4/3; \quad \acute{K}_s = (3/4)(\pi^2/9)^{\frac{1}{3}}\hbar c \simeq 3\hbar c/4, \qquad (3)$$

for the ultra-relativistic limit [13,14]. The degenerate pressure only depends on the number density of the plasma species but not on their temperature [13,14]. For the stable configuration of the DQPS, the outward directional degenerate pressure is counter-balanced by the inward gravitational pressure.

Mamun [11] first brought the idea of degenerate electron energy and the corresponding wave speed and wavelength by considering a cold DQPS containing inertialess degenerate electrons, inertial non-degenerate light nuclei, and stationary heavy nuclei, and showed that the degenerate pressure-driven nucleus-acoustic waves propagating in such a DQPS totally disappears if the degenerate pressure of the electrons is neglected. Mannan [15] investigated three-dimensional cylindrical waves in a self-gravitating DQPS and found that the considered plasma system supports both positive and negative electrostatic potentials and the amplitude, width, and speed are significantly modified by the effects of degenerate plasma species.

The electrostatic shock wave profile, which may arise due to the Landau damping and kinematic viscosity of the medium, is governed by the Burgers equation [16–19]. Atteya et al. [16] examined the ion-acoustic (IA) shock waves (IASHWs) in DQPS, and reported that the amplitude of the positive shock profile increases with the increase in electron number density. Abdelwahed et al. [17] investigated IASHWs in non-thermal plasma, and found that the steepness of the shock profile decreases with ion kinematic viscosity.

The external magnetic field has been considered to investigate the electrostatic shock [20–22] and solitary [23,24] waves in plasmas. Hossen et al. [22] examined the IASHWs in the presence of an external magnetic field, and highlighted that the amplitude of IASHWs increases when increasing the angle between the wave propagation vector and the direction of the external magnetic field (via $\delta$). Shaukat [23] studied IA solitary waves in degenerate magneto-plasma. Ashraf et al. [24] observed that the amplitude of the electrostatic shock wave increases with the oblique angle.

Recently, Islam et al. [14] investigated envelope solitions in a three-component DQPS containing relativistically degenerate electrons, positively charged heavy and light ions. To the best knowledge of the authors, no attempt has been made to study IASHWs in a magnetized DQPS with positively charged non-relativistic heavy and light ions, and ultra-relativistically degenerate electrons. Therefore, the aim of our present investigation is to derive the Burgers equation and by employing its shock solution, we numerically analyze the IASHWs in a magnetized DQPS.

The manuscript is organized in the following way: the governing equations are described in Section 2. The derivation of the Burgers equation and its shock solution are demonstrated in Section 3. The results and discussion are presented in Section 4. The conclusion is provided in Section 5.

## 2. Model Equations

We consider a magnetized DQPS consisting of inertial positively charged non-relativistic heavy ions (mass $m_1$; charge $q_1 = +eZ_1$; number density $N_1$; pressure $P_1$), positively charged non-relativistic light ions (mass $m_2$; charge $q_2 = +eZ_2$; number density $N_2$; pressure $P_2$), and inertialess ultra-relativistically degenerate electrons (mass $m_e$; charge $-e$; number density $N_e$; pressure $P_e$); where $Z_1$ ($Z_2$) is the charge state of the heavy (light) ion.

We also assume a uniform external magnetic field **B** in the direction of $z$-axis ($\mathbf{B} = B_0\hat{z}$). The propagation of IASHWs is governed by the following equations:

$$\partial_T N_1 + \tilde{\nabla} \cdot (N_1 U_1) = 0, \tag{4}$$

$$\partial_T U_1 + (U_1 \cdot \tilde{\nabla})U_1 = (Z_1 e B_0/m_1)(U_1 \times \hat{z})$$
$$+\tilde{\eta}\tilde{\nabla}^2 U_1 - (Z_1 e/m_1)\tilde{\nabla}\tilde{\Phi} - (1/m_1 N_1)\tilde{\nabla}P_1, \tag{5}$$

$$\partial_T N_2 + \tilde{\nabla} \cdot (N_2 U_2) = 0, \tag{6}$$

$$\partial_T U_2 + (U_2 \cdot \tilde{\nabla})U_2 = (Z_2 e B_0/m_2)(U_2 \times \hat{z})$$
$$+\tilde{\eta}\tilde{\nabla}^2 U_2 - (Z_2 e/m_2)\tilde{\nabla}\tilde{\Phi} - (1/m_2 N_2)\tilde{\nabla}P_2, \tag{7}$$

$$\tilde{\nabla}^2\tilde{\Phi} = 4\pi e(N_e - Z_2 N_2 - Z_1 N_1), \tag{8}$$

where $U_1$ ($U_2$) is the fluid speed of heavy (light) ion; $\tilde{\Phi}$ is the electrostatic wave potential; and $\tilde{\eta}$ is the kinematic viscosity for heavy and light ions, and for simplicity, we have assumed $\tilde{\eta} \simeq \tilde{\eta}_1/m_1 N_1 \simeq \tilde{\eta}_2/m_2 N_2$. The equation for the degenerate electron can be expressed as

$$\tilde{\nabla}\tilde{\Phi} - (1/eN_e)\tilde{\nabla}P_e = 0. \tag{9}$$

Now, we introduce the normalizing parameters as follows: $n_1 \to N_1/n_{10}$; $n_2 \to N_2/n_{20}$; $n_e \to N_e/n_{e0}$; $u_1 \to U_1/C_1$; $u_2 \to U_2/C_1$; $\phi \to e\tilde{\Phi}/m_e c^2$; $t \to T/\omega_{p1}^{-1}$; $\nabla \to \tilde{\nabla}/\lambda_{D1}$; $\eta = \tilde{\eta}/\omega_{p1}\lambda_{D1}^2$ (where $C_1 = (Z_1 m_e c^2/m_1)^{1/2}$; the plasma frequency $\omega_{p1}^{-1} = (m_1/4\pi Z_1^2 e^2 n_{10})^{1/2}$; the Debye length $\lambda_{D1} = (m_e c^2/4\pi Z_1 e^2 n_{10})^{1/2}$). At equilibrium, the quasi-neutrality condition can be written as $n_{e0} \simeq Z_1 n_{10} + Z_2 n_{20}$. By using these normalizing parameters, Equations (4)–(8) can be expressed as

$$\partial_t n_1 + \nabla \cdot (n_1 u_1) = 0, \tag{10}$$

$$\partial_t u_1 + (u_1 \cdot \nabla)u_1 = -\nabla\phi + \Omega_{c1}(u_1 \times \hat{z})$$
$$-(\mu_1 \acute{K}_1/n_1)\nabla n_1^\alpha + \eta\nabla^2 u_1, \tag{11}$$

$$\partial_t n_2 + \nabla \cdot (n_2 u_2) = 0, \tag{12}$$

$$\partial_t u_2 + (u_2 \cdot \nabla)u_2 = -\mu_2\nabla\phi + \mu_2\Omega_{c1}(u_2 \times \hat{z})$$
$$-(\mu_1 \acute{K}_2/n_2)\nabla n_2^\alpha + \eta\nabla^2 u_2, \tag{13}$$

$$\nabla^2\phi = (1+\mu_4)n_e - \mu_4 n_2 - n_1, \tag{14}$$

where the plasma parameters are: $\Omega_{c1} = \omega_{c1}/\omega_{p1}$ (where $\omega_{c1} = Z_1 e B_0/m_1$); $\mu_1 = m_1/Z_1 m_e$; $\mu_2 = Z_2 m_1/Z_1 m_2$; $\mu_3 = n_{e0}/Z_1 n_{10}$; $\mu_4 = Z_2 n_{20}/Z_1 n_{10}$; $K_1 = n_{10}^{\alpha-1}\acute{K}_1/m_1 c^2$; $K_2 = n_{20}^{\alpha-1}\acute{K}_2/m_2 c^2$ and $\gamma = \alpha = 5/3$ (for the non-relativistic limit). Now, by normalizing and integrating Equation (9), the number density of the inertialess electrons can be obtained in terms of electrostatic potential $\phi$ as

$$n_e = [1 + (\gamma_e - 1)\phi/K_3\gamma_e]^{1/(\gamma_e-1)}, \tag{15}$$

where $K_3 = n_{e0}^{\gamma_e-1}\acute{K}_e/m_e c^2$ and $\gamma = \gamma_e = 4/3$ (for the ultra-relativistic limit). Now, expanding the right hand side of Equation (15) and substituting in Equation (14), we can write

$$\nabla^2\phi + n_1 + \mu_4 n_2 = 1 + \mu_4 + \sigma_1\phi + \sigma_2\phi^2 + \cdots \tag{16}$$

where $\sigma_1 = [(\mu_4 + 1)/\alpha K_3]$ and $\sigma_2 = [(\mu_4 + 1)(2 - \gamma_e)/2(\alpha K_3)^2]$.

### 3. Derivation of the Burgers Equation

To study IASHWs, we derive the Burgers equation by introducing the stretched coordinates for independent variables as [22,23]

$$\xi = \epsilon(l_x x + l_y y + l_z z - v_p t), \tag{17}$$

$$\tau = \epsilon^2 t, \tag{18}$$

where $v_p$ is the phase speed and $\epsilon$ is a smallness parameter measuring the weakness of the dissipation ($0 < \epsilon < 1$). The $l_x$, $l_y$, and $l_z$ (i.e., $l_x^2 + l_y^2 + l_z^2 = 1$) are the directional cosines of the wave vector $k$ along $x$, $y$, and $z$-axes, respectively. The dependent variables can be expressed in power series of $\epsilon$ as [22]

$$n_1 = 1 + \epsilon n_1^{(1)} + \epsilon^2 n_1^{(2)} + \cdots, \tag{19}$$

$$n_2 = 1 + \epsilon n_2^{(1)} + \epsilon^2 n_2^{(2)} + \cdots, \tag{20}$$

$$u_{1x,y} = \epsilon^2 u_{1x,y}^{(1)} + \epsilon^3 u_{1x,y}^{(2)} + \cdots, \tag{21}$$

$$u_{2x,y} = \epsilon^2 u_{2x,y}^{(1)} + \epsilon^3 u_{2x,y}^{(2)} + \cdots, \tag{22}$$

$$u_{1z} = \epsilon u_{1z}^{(1)} + \epsilon^2 u_{1z}^{(2)} + \cdots, \tag{23}$$

$$u_{2z} = \epsilon u_{2z}^{(1)} + \epsilon^2 u_{2z}^{(2)} + \cdots, \tag{24}$$

$$\phi = \epsilon \phi^{(1)} + \epsilon^2 \phi^{(2)} + \cdots. \tag{25}$$

Now, by substituting Equations (17)–(25) into Equations (10)–(13) and (16), we obtain the lowest order in $\epsilon$ as Equations (A1)–(A8) (as given in Appendix A), along with the phase speed of IASHWs:

$$v_p = v_{p+} = l_z \sqrt{m_2 + \sqrt{(m_2^2 - 4m_1 m_3)/2m_1}}, \tag{26}$$

$$v_p = v_{p-} = l_z \sqrt{m_2 - \sqrt{(m_2^2 - 4m_1 m_3)/2m_1}}, \tag{27}$$

where $m_1 = \sigma_1$, $m_2 = 1 + \mu_2 \mu_4 - \alpha \sigma_1 \mu_1 K_2 - \alpha \sigma_1 \mu_1 K_1$, and $m_3 = \alpha \mu_1 K_2 + \alpha \mu_1 \mu_2 \mu_4 K_1 + \sigma_1 \alpha^2 \mu_1^2 K_1 K_2$.

The next higher order in $\epsilon$ gives a system of equations (given by Equations (A9)–(A13) in Appendix A). Solving this system with the help of (A1)–(A8) (as given in Appendix A), we finally obtain the Burgers equation as

$$\partial_\tau \Phi + A \Phi \partial_\xi \Phi = B \partial_{\xi\xi} \Phi, \tag{28}$$

where $\Phi = \phi^{(1)}$ for simplicity. In Equation (28), the nonlinear coefficient $A$ and dissipative coefficient $B$ are, respectively, given by

$$A = P(Q + R - 2\sigma_2), \tag{29}$$

$$B = \frac{\eta}{2}, \tag{30}$$

where

$P = [(v_p^2 - \alpha \mu_1 l_z^2 K_1)^2 (v_p^2 - \alpha \mu_1 l_z^2 K_2)^2]/2v_p l_z^2 [v_p^4 (1 + \mu_2 \mu_4) + \alpha^2 \mu_1^2 l_z^4 (K_2^2 + \mu_2 \mu_4 K_1^2) - M]$,

$M = 2\alpha \mu_1 l_z^2 v_p^2 (K_2 + K_1 \mu_2 \mu_4)$,

$Q = l_z^4 \{3v_p^2 + \mu_1 l_z^2 K_1 \alpha(\alpha - 2)\}/(v_p^2 - \alpha \mu_1 l_z^2 K_1)^3$,

$R = \mu_2^2 \mu_4^4 l_z^4 \{3v_p^2 + \alpha \mu_1 \mu_2 l_z^2 K_2(\alpha - 2)\}/(v_p^2 - \alpha \mu_1 l_z^2 K_2)^3$,

Now, we look for the stationary shock wave solution of this Burgers equation by considering $\zeta = \xi - U_0 \tau'$ and $\tau = \tau'$ (where $U_0$ is the speed of the shock waves in the reference frame). These allow us to write the stationary shock wave solution as [22,25,26]

$$\Phi = \Phi_0 \left[ 1 - \tanh \left( \frac{\zeta}{\Delta} \right) \right], \tag{31}$$

where the amplitude $\Phi_0$ and width $\Delta$ are given by

$$\Phi_0 = \frac{U_0}{A}, \quad \text{and} \quad \Delta = \frac{2B}{U_0}. \tag{32}$$

It is clear from Equations (31) and (32) that the IASHWs exist, which are formed due to the balance between nonlinearity and dissipation, because $B > 0$ and the IASHWs with $\Phi > 0$ ($\Phi < 0$) exist if $A > 0$ ($A < 0$) because $U_0 > 0$.

## 4. Results and Discussion

Our present investigation is valid for white dwarfs and neutron stars in which both non-relativistic positively charged heavy ions (e.g., $^{56}_{26}$Fe [4], $^{85}_{37}$Rb [5], $^{96}_{42}$Mo [5]), and light ions (e.g., $^{1}_{1}$H [6,7], $^{4}_{2}$He [8], $^{12}_{6}$C [9,10]), and ultra-relativistically degenerate electrons exist. For numerical analysis, we considered $Z_1 = 20 \sim 60$, $Z_2 = 1 \sim 12$, $n_{10} = 1 \times 10^{29}$ cm$^{-3}$ $\sim$ $9 \times 10^{29}$ cm$^{-3}$, $n_{20} = 2 \times 10^{30}$ cm$^{-3}$ $\sim 8 \times 10^{30}$ cm$^{-3}$, and $n_{e_0} = 10^{32}$ cm$^{-3}$ $\sim 10^{34}$ cm$^{-3}$. The IASHWs are governed by the Burgers equation (28), and the positive (negative) shock potential can exist corresponding to the limit of $A > 0$ ($A < 0$). The variation of $A$ with $\mu_4$ can be seen from Figure 1 (left panel), and it is clear from this figure that our plasma model only supports positive shock potential under the consideration of both non-relativistic positively charged heavy and light ions (i.e., $\alpha = 5/3$), and ultra-relativistically degenerate electrons (i.e., $\gamma_e = 4/3$).

The parameter $\delta$ reveals the angle between the direction of the wave propagation and the direction of the external magnetic field, and the effects of $\delta$ on the formation of IASHWs can be seen in Figure 1 (right panel). When the oblique angle ($\delta$) increases, the magnetic effect becomes more significant, and therefore the amplitude of the shock wave increases, and this result agrees with the result of Hossen et al. [22].

Figure 2 (left panel) illustrates the effects of the non-relativistic heavy and light ion's kinematic viscosity on the positive potential (i.e., $\Phi > 0$) under consideration of $A > 0$. It is really interesting that the steepness of the shock profile decreases with an increase in the value of the non-relativistic heavy and light ion's kinematic viscosity, but the amplitude of shock profile is not affected by the kinematic viscosity of ions, and this result agrees with the previous work of Abdelwahed et al. [17].

The variation of IASHWs with electron number density ($n_{e0}$) under the consideration of both non-relativistic positively charged heavy and light ions (i.e., $\alpha = 5/3$), and ultra-relativistically degenerate electrons (i.e., $\gamma_e = 4/3$) can be observed in Figure 2 (right panel). It is clear from this figure that as we increase the electron number density, the amplitude of the IASHWs associated with $\Phi > 0$ (i.e., $A > 0$) increases. So, the ultra-relativistic electrons enhance the amplitude of the IASHWs in a magnetized DQPS with non-relativistic positively charged heavy and light ions, and ultra-relativistically degenerate electrons.

The effects of the charge state of non-relativistic heavy and light ions species on the formation of IASHWs in a magnetized DQPS can be seen in the left panel and right panel of Figure 3, respectively. It is obvious from these figures that the charge state of both non-relativistic heavy and light ion species enhances the amplitude of IASHWs associated with $\Phi > 0$ (i.e., $A > 0$) under the consideration of $\alpha = 5/3$ and $\gamma_e = 4/3$. Physically, both non-relativistic heavy and light ion species, due to both being positively charged, play the same role in the dynamics of magnetized DQPS as well as the configuration of IASHWs. Similarly, the number density of the non-relativistic heavy and light ion species can play a

significant role in the formation of IASHWs. It is clear from the figures in both panels of Figure 4 that the amplitude of the IASHWs associated with $\Phi > 0$ (i.e., $A > 0$) and under the consideration of $\alpha = 5/3$ and $\gamma_e = 4/3$ increases with the number density of both non-relativistic heavy and light ion species.

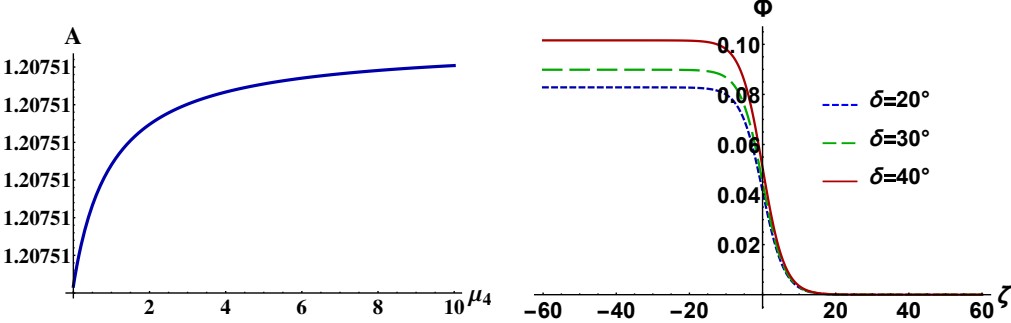

**Figure 1.** Plot of the nonlinear coefficient $A$ vs. $\mu_4$ (**left panel**) and $\Phi$ vs $\zeta$ for different values of $\delta$ (**right panel**) when $\alpha = 5/3$, $\eta = 0.3$, $\gamma_e = 4/3$, $\delta = 20°$, $Z_1 = 37$, $Z_2 = 6$, $n_{10} = 10^{29}$ cm$^{-3}$, $n_{20} = 10^{30}$ cm$^{-3}$, $n_{e_0} = 10^{33}$ cm$^{-3}$, $U_0 = 0.05$, and $v_p = v_{p+}$.

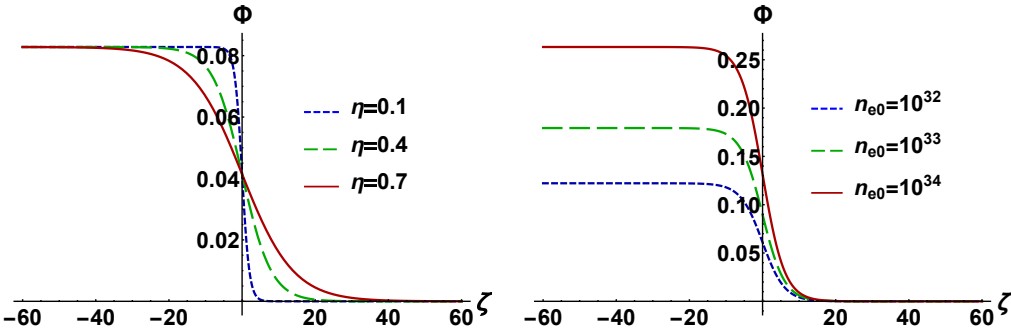

**Figure 2.** Plot of $\Phi$ vs. $\zeta$ for different values of $\eta$ (**left panel**) and for different values of $n_{e_0}$ (**right panel**) when $\alpha = 5/3$, $\delta = 20°$, $\gamma_e = 4/3$, $\eta = 0.3$, $Z_1 = 37$, $Z_2 = 6$, $n_{10} = 10^{29}$ cm$^{-3}$, $n_{20} = 10^{30}$ cm$^{-3}$, $n_{e_0} = 10^{33}$ cm$^{-3}$, $U_0 = 0.05$, and $v_p = v_{p+}$.

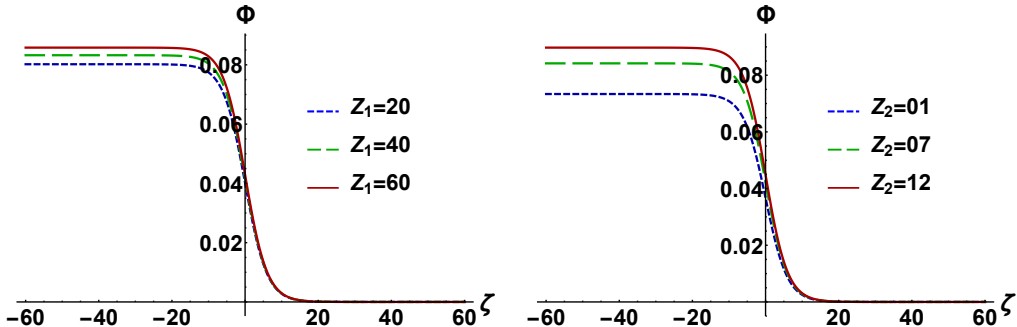

**Figure 3.** Plot of $\Phi$ vs. $\zeta$ for different values of $Z_1$ (**left panel**) and for different values of $Z_2$ (**right panel**) when $\alpha = 5/3$, $\delta = 20°$, $\eta = 0.3$, $\gamma_e = 4/3$, $Z_1 = 37$, $Z_2 = 6$, $n_{10} = 10^{29}$ cm$^{-3}$, $n_{20} = 10^{30}$ cm$^{-3}$, $n_{e_0} = 10^{33}$ cm$^{-3}$, $U_0 = 0.05$, and $v_p = v_{p+}$.

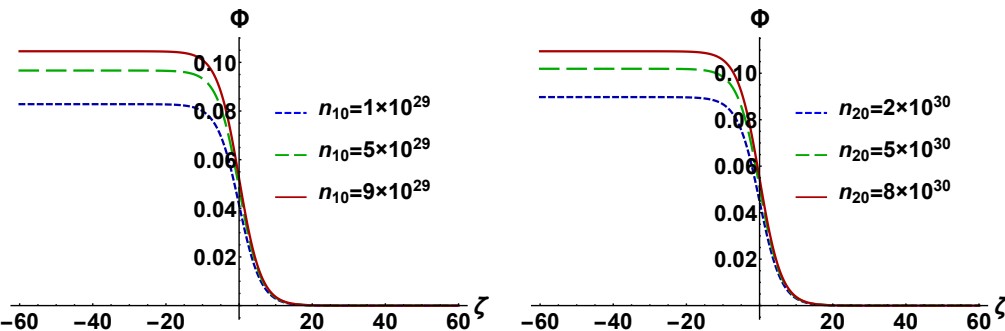

**Figure 4.** Plot of $\Phi$ vs. $\zeta$ for different values of $n_{10}$ (**left panel**) and for different values of $n_{20}$ (**right panel**) when $\alpha = 5/3$, $\delta = 20°$, $\eta = 0.3$, $\gamma_e = 4/3$, $Z_1 = 37$, $Z_2 = 6$, $n_{10} = 10^{29}$ cm$^{-3}$, $n_{20} = 10^{30}$ cm$^{-3}$, $n_{e_0} = 10^{33}$ cm$^{-3}$, $U_0 = 0.05$, and $v_p = v_{p+}$.

## 5. Conclusions

We investigated the fundamental characteristics of IASHWs in a magnetized DQPS with inertial non-relativistic positively charged heavy and light ions, inertialess ultra-relativistically degenerate electrons. The reductive perturbation method [27–31] was employed to derive the Burgers equation. The results found from the present study can be pinpointed as follows:

- The plasma model only supports positive shock potential under the consideration of both non-relativistic positively charged heavy and light ions (i.e., $\alpha = 5/3$), and ultra-relativistically degenerate electrons (i.e., $\gamma_e = 4/3$);
- The increasing number density of ultra-relativistic electrons enhances the amplitude of the IASHWs;
- The increasing charge state and number density of the non-relativistic heavy and light ion species enhance the amplitude of the IASHWs associated with $\Phi > 0$ (i.e., $A > 0$);
- The steepness of the shock profile is decreased with the increasing kinematic viscosity ($\eta$) of ions;
- The amplitude of the shock profile is found to increase as the oblique angle increases.

It may be noted here that it is really important to include the exchange and correlation effects of plasma species [32,33] and the self-gravitational effects of the DQPS in the governing equations, but this is beyond the scope of our present work. It is also important to mention that the Bohm potential arises due to the effect of quantum diffraction or quantum tunneling, and that in the case of a long wavelength, the Fermi temperature term may dominate over the Bohm potential term; thus, we neglect the Bohm potential term compared to the Fermi temperature term in the equation of motion [34,35]. Thus, we neglected the Bohm potential in quantum plasma, as many published works by many authors did [34–38]. However, we are optimistic that the outcomes from our present investigation will be useful in understanding the propagation of IASHWs in white dwarfs [1–3] and neutron stars [1–3] in which the non-relativistic positively charged heavy and light ions, and ultra-relativistically degenerate electrons exist.

**Author Contributions:** All authors contributed equally to complete this work. All authors have read and agreed to the published version of the manuscript.

**Funding:** This research received an external funding from "UGC research project 2018–2019".

**Institutional Review Board Statement:** Not applicable.

**Informed Consent Statement:** Not applicable.

**Data Availability Statement:** The data that support the findings of this study are available from the corresponding author upon reasonable request.

**Acknowledgments:** Authors would like to acknowledge "UGC research project 2018–2019" for their financial supports to complete this work.

**Conflicts of Interest:** The authors declare no conflict of interest.

## Appendix A. First-Order and Second-Order Perturbation Terms

By collecting the terms containing the coefficients of $\epsilon$ from Equations (10)–(13) and (16) we obtain the first-order equations as:

$$n_1^{(1)} = [l_z^2/(v_p^2 - \alpha\mu_1 l_z^2 K_1)]\phi^{(1)}, \tag{A1}$$

$$u_{1z}^{(1)} = [v_p l_z/(v_p^2 - \alpha\mu_1 l_z^2 K_1)]\phi^{(1)}, \tag{A2}$$

$$n_2^{(1)} = [\mu_2 l_z^2/(v_p^2 - \alpha\mu_1 l_z^2 K_2)]\phi^{(1)}, \tag{A3}$$

$$u_{2z}^{(1)} = [\mu_2 v_p l_z/(v_p^2 - \alpha\mu_1 l_z^2 K_2)]\phi^{(1)}. \tag{A4}$$

The $x$ and $y$-components of the first-order momentum equations can be manifested as:

$$u_{1x}^{(1)} = [-l_y v_p^2/\Omega_{c1}(v_p^2 - \alpha\mu_1 l_z^2 K_1)]\partial_\xi\phi^{(1)}, \tag{A5}$$

$$u_{1y}^{(1)} = [l_x v_p^2/\Omega_{c1}(v_p^2 - \alpha\mu_1 l_z^2 K_1)]\partial_\xi\phi^{(1)}, \tag{A6}$$

$$u_{2x}^{(1)} = [-l_y v_p^2/\Omega_{c1}(v_p^2 - \alpha\mu_1 l_z^2 K_2)]\partial_\xi\phi^{(1)}, \tag{A7}$$

$$u_{2y}^{(1)} = [l_x v_p^2/\Omega_{c1}(v_p^2 - \alpha\mu_1 l_z^2 K_2)]\partial_\xi\phi^{(1)}. \tag{A8}$$

Finally, by collecting the next higher-order terms of Equations (10)–(13) and (16) we obtain the following second-order equations:

$$\partial_\tau n_1^{(1)} - v_p\partial_\xi n_1^{(2)} + l_x\partial_\xi u_{1x}^{(1)} + l_y\partial_\xi u_{1y}^{(1)} + l_z\partial_\xi u_{1z}^{(1)} + l_z\partial_\xi\left(n_1^{(1)}u_{1z}^{(1)}\right) = 0, \tag{A9}$$

$$\partial_\tau u_{1z}^{(1)} - v_p\partial_\xi u_{1z}^{(2)} + l_z u_{1z}^{(1)}\partial_\xi u_{1z}^{(1)} + l_z\partial_\xi\phi^{(2)} - \eta\partial_{\xi\xi}u_{1z}^{(1)} + \alpha\mu_1 l_z K_1\left[\partial_\xi n_1^{(2)} + [(\alpha-2)/2]\partial_\xi n_1^{(1)2}\right] = 0, \tag{A10}$$

$$\partial_\tau n_2^{(1)} - v_p\partial_\xi n_2^{(2)} + l_x\partial_\xi u_{2x}^{(1)} + l_y\partial_\xi u_{2y}^{(1)} + l_z\partial_\xi u_{2z}^{(2)} + l_z\partial_\xi\left(n_2^{(1)}u_{2z}^{(1)}\right) = 0, \tag{A11}$$

$$\partial_\tau u_{2z}^{(1)} - v_p\partial_\xi u_{2z}^{(2)} + l_z u_{2z}^{(1)}\partial_\xi u_{2z}^{(1)} + \mu_2 l_z\partial_\xi\phi^{(2)} - \eta\partial_{\xi\xi}u_{2z}^{(1)} + \alpha\mu_1 l_z K_2\left[\partial_\xi n_2^{(2)} + [(\alpha-2)/2]\partial_\xi n_2^{(1)2}\right] = 0, \tag{A12}$$

$$\mu_4 n_2^{(2)} + n_1^{(2)} = \sigma_1\phi^{(2)} + \sigma_2\phi^{(1)2}. \tag{A13}$$

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
