# Peer review of "Electrostatic Ion-Acoustic Shock Waves in a Magnetized Degenerate Quantum Plasma"

_plasma, doi:10.3390/plasma4030031_

Round 1

Reviewer 1 Report

The paper is an interesting theoretical analysis of shock waves in magnetized degenerate plasmas. The authors present a fairly complete picture by including non-relativistic heavy ions, non-relativistic light ions, and ultra-relativistic degenerate electrons. The authors are able to show that their Euler like fluid equations coupled to the Poisson lead to a Burgers equation. The latter supporting discontinuous shock like solutions. One suggestion I have since the paper involves a lengthy list of equations. The paper might be easier to read if some of the material on the derivation of the Burgers equation is moved to an appendix. Section 3 could be a summary of the derivation with more discussion of the physics.

Reviewer 2 Report

In this paper, the authors investigate ion-acoustic shock waves in a magnetized plasma composed of inertialess ultra-relativistically degenerate electrons, inertial non-relativistic positively charged heavy
and light ions. They adopt a hydrodynamic model and employ the standard perturbation technique to derive the corresponding Burger equation and discuss its solution. In my opinion, the overall results may appeal to the plasma physics community. However, I have a few concerns and remarks about the manuscript.

- My main concern is that, although considering a quantum plasma, it is only the pressure that is obtained quantum-mechanically (i.e., Eq. (1)). The dynamics, instead, is treated on classical grounds, as indicated by the Euleur (momentum balance) equations (Eqs. (5) and (7)). Why is the effect of the quantum (Bohm) potential not considered in these equations? Are the authors considering low enough densities such that the quantum potential can be neglected but sufficiently low temperatures, so that degeneracy effects dominate? Are these conditions realizable in physical or astrophysical scenarios? This must be explained.

- The authors may enhance the quality of the introduction by presenting concrete physical situations where such a model can be found
meaningful (e.g., in white dwarfs or neutron stars, as they claim), with proper reference to the recent literature. Also, in the conclusion, where they present a number of possible follow-ups to the paper, the authors may want to include exchange-correlation effects, that may be incorporated in a similar model (see for instance [Phys. Rev. E 88, 063105 (2013); Phys. Rev. E 88, 045101 (2013)]).

-  Overall, the manuscript is clearly written but contains a number of syntax errors. The authors should carefully revise the whole manuscript to make it 
free of language mistakes. 

If the authors consider my remarks with care, I will be happy to recommend the paper for publication.

Round 2

Reviewer 2 Report

All my remarks have been taken into account. I recommend the paper for publication.